# Impact of Milk Thistle (*Silybum marianum* [L.] Gaertn.) Seeds in Broiler Chicken Diets on Rearing Results, Carcass Composition, and Meat Quality

**DOI:** 10.3390/ani11061550

**Published:** 2021-05-26

**Authors:** Alina Janocha, Anna Milczarek, Daria Pietrusiak

**Affiliations:** Institute of Animal Science and Fisheries, Faculty of Agrobioengineering and Animal Husbandry, Siedlce University of Natural Sciences and Humanities, Bolesława Prusa 14, 08-110 Siedlce, Poland; alina.janocha@uph.edu.pl (A.J.); dp335@stud.uph.edu.pl (D.P.)

**Keywords:** milk thistle, rations, broiler chicken, performance results, carcass value, meat quality

## Abstract

**Simple Summary:**

This study shows how introducing milk thistle seeds into broiler chicken feed rations affects rearing results (weight gain, intake and conversion of feed), carcass composition and meat quality (pH, color, water holding capacity), the chemical composition (basic components, fatty acids) and organoleptic properties (flavor, tenderness, palatability and juiciness) of meat. Based on the results the use of milk thistle seeds in broiler chicken starter/grower diets can be recommended in the amount of 0/2% or 2/3%, respectively. However, the introduction of *Silybum marianum* in starter and grower rations (over the whole rearing period) made it possible to obtain the highest body weight at the lowest feed conversion per body weight gain unit, without influencing muscularity and fattening grade, at the same time improving the meat’s value for health.

**Abstract:**

The studies aimed to evaluate the impact of milk thistle seeds in broiler chicken feed rations on rearing results, carcass composition and meat quality. The experiment involved 120 broiler chickens randomly allocated to three equinumerous groups (C, MT02, MT23). Each group was divided into five subgroups of eight chickens each. Over the first 21 days of life the birds were fed starter rations, and over the following 21 days received grower rations. Chicken starter/grower diets in groups MT02 and MT23 were supplemented with ground seeds of milk thistle in the amount of 0/2% (MT02) and 2/3% (MT23). It was demonstrated that *Silybum marianum* added to feed rations over the whole rearing period (group MT23) increased above 3% the birds’ body weight on rearing day 42. (*p* < 0.05) and decreased about 7% the feed conversion ratio (*p* < 0.05) in comparison to group C. No effect of feeding on the carcass composition was observed, including on muscularity and fattening grade, although diets containing milk thistle reduced (by 15% and 19% in group MT02 and MT23, respectively) the content of crude fat in chicken leg muscles (*p* < 0.05). The highest (*p* < 0.05) content of polyunsaturated fatty acids (PUFA) was determined in the breast (38.06%) and leg (37.63%) muscles of chicken receiving feed rations containing *Silybum marianum* throughout the rearing period. No effect of nutrition on the evaluated physical properties of muscles was observed, except on the decrease in lightness color (L*) and increase in values a* and C as well as a decrease of water holding capacity of the breast muscles. It was found that *Silybum marianum* in chicken diets had a positive effect on the evaluated meat flavor characteristics of the muscles. To sum up, based on the study results, including ground seeds of milk thistle in broiler chickens nutrition can be recommended in the amount of 2/3% in starter/grower diets, respectively.

## 1. Introduction

The European Union’s prohibition of using antibiotics in feeds as growth stimulants made breeders and feed producers look for new nutritional solutions. Apart from probiotics, prebiotics, organic acids, antioxidants, and feed enzymes, various preparations of plant origin—so-called phytobiotics—have raised more and more interest [1,2,3,4,5]. Hippenstiel et al. [6] and Tavakolinasab et al. [7] report that herbs can be used in various forms: fresh or dried, infusion, brew, extract, essential oils and macerates.

Herbs and herbal preparations contain active biological ingredients, i.e.,: essential oils, tannins, glycosides, flavonoids, terpenes, mucilages, and organic acids [5,8,9,10]. Biological active components of herbal preparations display multi-directional activity: anti-stress, anti-bacterial, anti-virus, and anti-fungicidal [11,12]. In addition, they enhance the excretion of digestive enzymes increasing the appetite of animals [8,13,14]. They are believed “natural” and “safe” [12]. A positive impact of medicinal plants on the microflora of the digestive tract and productive performance of animals was demonstrated [4,15,16,17]. In the opinion of Gregačević et al. [18] phytogenic feed additives stimulate the immunological system, whereas the resistant stimulating effect of these additives is based on an increased activity of lymphocytes, macrophages, and cells.

Currently, milk thistle (*Silybum marianum* [L.] Gaertn) is increasingly popular in animal nutrition [9,19,20,21]. Milk thistle seeds have a different content of protein (160–300 g/kg) containing a lot of exogenous amino acids, which is confirmed by EAAI—the essential amino acid index (60.32) [22,23,24]. Crude fat from milk thistle seeds has an advantageous composition of fatty acids with oleic and linoleic acid predominating [22,23,25,26,27]. A high content of cellulose and lignin fraction in the diets of monogastric animals can pose certain limitations [28]. The main biological active component of milk thistle seeds is silymarin accounting for 1.5–3% [24,29]. Silymarin is composed of flavonolignans consisting of: silibinin (60%), isosilibinin (5%), silidianin (20%) and silicristin (10%), of which silibinin has the strongest effect [29,30,31,32]. Silymarin perfectly protects the liver from toxic agents and helps regenerate it [30,33]. In addition, it has an anti-inflammatory effect, inhibiting the migration of neutrophils and fostering the formation of prostaglandins [34,35,36]. Due to its medicinal properties, silymarin is preferred by veterinary doctors as a natural medicine and recommended for intensive animal production [7,37,38,39,40]. Therefore, milk thistle can be widely used for animal feeding and in veterinary medicine. However, the results of studies [20,41,42,43,44] differ. Šťastník et al. [43] demonstrated that using 5% or 15% of milk thistle oil cake in the diets of broiler chickens can decrease their weight gain. Similarly, Gharahveysi [44] introducing 0.3% or 3% ground milk thistle into the birds’ diets found a reduced feed intake and body weight of chickens. However, Muhammad et al. [42] and Ahmad et al. [20] showed an increase in body weight and feed conversion ratio in chickens fed diets containing 10 and 15 g/kg of milk thistle, respectively. Tedesco et al. [31] and Mojahedtalab et al. [45] showed a positive effect of supplementing broiler chicken feed rations with silymarin-phospholipid complex or silymarin on the performance results. Similarly, the post-slaughter results of birds fed diets containing *Silybum marianum* do not unanimously show how it affects the carcass composition and meat quality, since available literature lacks an evaluation the physico-chemical and organoleptic characteristics of meat [7,43,46,47,48,49].

The aim of the studies was to assess the effect of feeding different amounts of milk thistle seeds at different productive stages (starter and grower) on broiler productive performance, carcass composition, meat quality and sensory properties.

## 2. Materials and Methods

### 2.1. Experiment Design

The feeding experiment involved 120 Ross 308 sexed broiler chickens randomly allocated to three equinumerous groups (C, MT02, MT23). Each group was divided into five subgroups of eight chickens (4 males and 4 females) each. The birds were reared over a 42–day cycle in metal cages under standard microclimate conditions with unlimited access to water (nipple drinkers) and feed. A free feeding (ad libitum) scheme was used. Throughout the rearing period the birds were exposed to 24–hour electric lighting. In the first experimental week ambient temperature was 32 °C, and was then decreased every week (every 7 days) by 1–2 °C until it reached 21–23 °C in the last rearing week. In the first rearing period, i.e., until day 21, the birds were fed complete bulk starter feed rations and from day 22 to 42—with grower feed rations. All loose feed rations were based on wheat, soybean cake (non–GMO), soybean oil and mineral and vitamin additives. Chicken starter/grower diets in experimental groups (MT02 and MT23) were supplemented with ground seeds of milk thistle in the amount of 0/2% (MT02) and 2/3% (MT23). The nutrient content of the basal diet was calculated on the basis of the chemical composition of raw feedstuffs, and the metabolizable energy value was in line with equations from the European Tables [50]. The nutritional value of rations was calculated according to nutritional recommendations and indicated in Table 1.

During the growth experiment, the birds’ body weight was controlled on day 1, 21 and 42 along with the intake of feed in respective rearing periods. The results were used to calculate weight gain and feed conversion (FCR) per weight gain unit.

On the 42nd day of the birds’ life, ten birds (five males and five females) with a body weight representative of a specific group and sex were selected from each group and slaughtered. Next, the carcasses were cooled over 24 h at a temperature of 4 °C. To calculate the dressing percentage, the weight of cooled carcasses was determined and they were subject to simplified dissection analysis using a procedure described by Ziołecki and Doruchowski [51]. During dissection, samples of breast and leg muscles were taken for evaluating their physico-chemical and organoleptic characteristics.

### 2.2. Chemical Composition Evaluation of Milk Thistle and Muscles

The samples for analysis were collected according to applicable requirements [52]. The dry matter, total ash, crude protein, and crude fat contents were described by the AOAC [53] according method number: dry matter (930.15), total ash (942.05), crude protein (990.03), crude fat (991.36) and crude fiber (978.10). The gross energy of milk thistle was determined using an Oxygen Bomb Calorimeter [54]. The number of nitrogen-free extractives (NFE) was calculated from the formula:NFE = dry matter − (crude protein + total ash + crude fat + crude fiber) 

The fatty acid profile in milk thistle and in muscles was determined by gas chromatography [55]. Fatty acid analysis was made with gas chromatography (GC) using gas chromatograph (GCMS-QP210 Ultra, Shimadzu, Kyoto, Japan) with capillary column and flame-ionization detection and helium as the carrier gas. The initial oven temperature was 140 °C for 1 min, thereafter increased by 20 °C/min to 200 °C and held for 20 min and increased by 5 °C/min to 220 °C held for 25 min. The injector was heated to 250 °C and the detector to 270 °C. FAME standards (Supelco 37 Component FAME Mix) were used to identify the fatty acids present in the samples. Based on the percentage (% of the total) of fatty acids, we calculated the atherogenic (AI) and thrombogenic (TI) indexes, as well as the hypocholesterolemic-to-hypercholesterolemic fatty acids ratio (HH) according to Ulbricht and Southgate [56] and Santos-Silva et al. [57]:AI = (C12:0 + 4 × C14:0 + C16:0)/[ΣMUFA + Σ(n–6) + Σ(n–3)] 
TI = (C14:0 + C16:0 + C18:0)/[0.5 × ΣMUFA + 0.5 × Σ(n–6) + 3 × Σ(n–3) + Σ(n–3)/Σ(n–6)]
HH = [(C18:1n–9 + C18:2n–6 + C20:4n–6 + C18:3n–3 + C20:5n–3 + C22:5n–3 + C22:6n–3)/(C14:0 + C16:0)]

### 2.3. Physical Properties Evaluation of Muscles

The concentration of hydrogen ions (pH_15_ and pH_24_) in *pectoralis maior* and *iliotibialis* muscles was measured using a Testo 205 pH-meter with a dagger electrode. Fifteen minutes after the slaughter and after over 24 h of cooling the reaction (pH_15_ and pH_24_) was measured in muscles.

Water absorption expressed as water holding capacity (WHC) was determined by Grau and Hamm’s filter-paper press method described by Jurczak [58] based on the surface of meat juice on the filter-paper.

The color of breast muscles was determined using a Minolta Chroma Metters CR 300 (Konica Minolta Osaka, Japan) instrument according to the L, a*, b* system [59]. Two illuminant/observer combinations were applied, i.e., illuminant C (average day light) and standard observer 2° as well as illuminant D65 (day light) and standard observer 10°, recommended for measurements of meat color [60]. In the used measuring system L denotes psychometric color saturation that is a spatial vector. On the other hand, a* and b* are trichromatic coordinates, where a* as a positive value corresponds to red, and as a negative value—green; in turn, positive b* corresponds to yellow, and negative b*—blue. The color parameters a* and b* were used to calculate chroma (C) and hue (H) with formulas used by [61].

### 2.4. Organoleptic Properties of Muscles

The organoleptic properties of breast and thigh muscles were evaluated on a five-point scale after cooking in a 0.8% NaCl solution up to a temperature of 80 °C in the geometric center of the sample. The meat to water ratio was 1:2. The flavor of muscles in terms of palatability, flavor, juiciness and tenderness was evaluated by a group of eight trained people [62,63].

### 2.5. Statistical Analysis

The results were elaborated by statistical methods using one-way analysis of variance, according to the following mathematical model:Y_ik_ = µ + a_i_ + e_ik_
where: Y_ik_—trait level,µ—total mean,a_i_—effect of treatment,e_ik_—error.

The significance of differences between mean values was verified using Tukey test at the significance level α ≤ 0.05. The results were elaborated using STATISTICA PL 13.1 software [64].

## 3. Results

The evaluated seeds of milk thistle (*Silybum marianum* [L.] Gaertn.) contained 219 g/kg of total protein and 238 g/kg of crude fat with an advantageous fatty acid composition (Table 2).

The seeds contained only 19.08% of SFA (saturated fatty acids). The content of MUFA (monounsaturated fatty acids) exceeded 24.60% and PUFA (polyunsaturated fatty acids) were predominant—54.99%. Among all monounsaturated acids, milk thistle contained the largest share of oleic acid—23.59%, and the polyunsaturated acids profile was dominated by linoleic acid—54.64%. As a result, nearly 85% of total fatty acids (FA) in milk thistle seeds were neutral and hypocholesterolemic acids (DFA).

The inclusion of milk thistle seeds in starter/grower diets in the amount 2/3% (MT23 group) significantly (by more than 3%) increased the body weight of broilers on the 42nd day of rearing in comparison to group C (Table 3).

Milk thistle in starter diets (group MT23) significantly decreased feed conversion in comparison to other groups. In the second rearing period and throughout the entire rearing period—chickens fed diets with milk thistle showed more efficient (*p* ≤ 0.05) conversion of feed. Differences in FCR between groups MT02 and MT23 and group C were 3% and 7%, respectively, throughout the rearing period. The type of feed rations used did not affect the post-slaughter performance, except the share of drumstick muscles (Table 4).

The presence of ground seeds of *Silybum marianum* in starter and grower diets (MT02 group) significantly increased the share of drumstick muscles in cold carcass compared to muscles of birds from MT23 group (9.03% vs. 8.13%).

The nutrition used significantly affected the content of crude ash in breast muscles and the level of crude fat in leg muscles (Table 5).

The breast muscles of birds from group MT02 contained significantly less crude ash than the muscles of birds from group C. Milk thistle seeds in chicken diets (groups MT02 and MT23) contributed to decreasing (*p* < 0.05) the content of crude fat in leg muscles.

Table 6 describes the composition and share of fatty acids in the lipid fraction of breast and leg muscles.

Feeding chickens with feed rations containing 2% (starter) and 3% (grower) of *Silybum marianum* significantly increased the share of stearic acid C_18:0_ (classified as neutral and hypocholesterolemic acid) in leg muscles only. Milk thistle introduced into chicken diets increased the content of linoleic acid (C_18:2_) in both evaluated muscle types, but a higher content of this acid was found in the muscles of chickens from group MT23 in comparison to the control group. Breast muscles of birds receiving feed rations with milk thistle in both rearing periods (MT23 group) contained by about 42% more linoleic acid than in group C (*p* < 0.05). A high (*p* < 0.05) content of PUFA was determined in the breast (38.06%) and leg (37.63%) muscles of chicken receiving feed rations containing ground seeds of *Silybum marianum* throughout the birds’ rearing period in comparison to group C. In addition, the breast muscles of birds from group MT23 featured a lower (*p* < 0.05) ratio of n–6:n–3 fatty acids in comparison to chickens fed rations without the phytobiotic.

The introduction of *Silybum marianum* into chicken diets did not affect the evaluated physical properties (pH, color, WHC) of thigh muscles, but it changed the color and WHC of breast muscles (Table 7).

Breast muscles of chickens from group MT02—fed with rations containing the evaluated phytobiotic at the grower stage only—were darker (*p* < 0.05). In addition, the use of milk thistle in the diets of birds intensified meat color toward red (a*) and increased chroma (C) of breast muscles compared to the C group. The water holding capacity (WHC) was lower (*p* < 0.05) in the breast muscles of chickens receiving diets containing *Silybum marianum*.

Adding ground seeds of milk thistle both in starter diets (20 g/kg) and grower diets (30 g/kg) and only in grower diets (20 g/kg) had a significant (*p* < 0.05) impact on the evaluated taste characteristics (flavor, tenderness, palatability, juiciness) of breast and thigh muscles, except on juiciness in breast muscles (*p* > 0.05) (Figure 1 and Figure 2).

Breast and thigh muscles of chickens fed with rations containing milk thistle in both rearing periods (group MT23) scored the highest.

## 4. Discussion

The content of total protein in the evaluated seeds of milk thistle (*Silybum marianum* [L.] Gaertn.) ranged from 161 to 250 g/kg as reported by many researchers [21,22,28,65]. In turn, a higher content of protein in milk thistle seeds (up to 30.09%), depending on the variety and origin, was found by Aziz et al. [24]. In the opinion of Grela et al. [21] and Aziz et al. [24] milk thistle seeds are not only a source of protein but also energy, which is corroborated by the level (up to 24.8%) of crude fat determining the energy value of the raw material. A lower (17.5–21.6%) content of fat in milk thistle seeds was demonstrated by Růžičková et al. [66], and a higher content (26.05–30.5%) of this component was found in the experiments by [25,26]. According to [25,26,67,68], the content of crude fat in milk thistle seeds depends on many factors such as: agricultural engineering, the environment, variety, and year of harvest. The determined content of crude fiber in the evaluated seeds amounted to 41.3 g/kg, which was below the range of 45.6–54.6 g/kg reported by Grela et al. [23].

An analysis of the fatty acids profile of *Silybum marianum* seeds showed a high level of unsaturated fatty acids, including linoleic acid (54.64% FA) and oleic acid (23.59% FA). Similar content of fatty acids was measured by Garaev et al. [69], Růžičková et al. [66] and Harrabi et al. [26], who in the evaluated samples of milk thistle seeds determined 50.58–66.4% of linoleic acid, 16.26–25.44% of oleic acid, 7.24–9.20% of palmitic acid and 3.56–5.92% of stearic acid. The above-mentioned higher shares of linoleic acid (64.4%) and oleic acid (26.38%) in the lipids of *Silybum marianum* seeds were found by Khan et al. [25], whereas a lower (45.36% and 39%) share of linoleic acid but a higher (31.58% and 36%) share of oleic acid was determined by Hasanloo et al. [70] and Majidi et al. [27], respectively.

Grela at al. [23] demonstrated a share of MUFA (24.98% vs. 24.60% FA) and that of PUFA (55.56% vs. 54.99% FA), which is similar to the level determined by the present authors. Wierzbowska et al. [68] found that the share of polyunsaturated fatty acids (PUFA), including linoleic acid, was lower than in own studies. In turn, Kralik et al. [47] noted a higher content (63.11%) of PUFA in milk thistle seeds. Big differences in the share of respective fatty acids in the seeds of *Silybum marianum* depending on the year of harvest were revealed by Sadowska et al. [28].

A positive impact on weight gain and feed conversion after introducing milk thistle into the feed rations for broiler chickens was observed by Tedesco et al. [31] and Muhammad et al. [42]. Mojahedtalab et al. [45] noted a linear improvement in body weight gain with a decreasing conversion of feed and an increasing content of silymarin in feed rations for broiler chickens. Similarly, Ahmad et al. [20] showed that overall body weight and feed conversion ratio were significantly (*p* < 0.5) higher for group MT–15 (rations 15 g/kg of milk thistle) compared with other experimental groups. In turn, Gharahveysi [44] found that the feed intake and body weight of chickens decreased as the share of milk thistle in their diet increased. Feed intake in diets containing 0.3% or 3% of *Silybum marianum* was 3.440 and 3.407 kg, respectively, and body weight was 1.810 and 1.793 kg, respectively. Other authors who demonstrated a reduction in the productivity of broiler chickens after including *Silybum marianum* in their diets were Suchý et al. [41], Kalantar et al. [71] and Šťastník et al. [43]. Suchý et al. [41] found that the use of 0.2% or 1% milk thistle seed expeller in feed rations for Ross 308 broilers resulted in decreasing weight gain and impaired feed conversion. Similarly, Kalantar et al. [71] showed that 0.5% *Silybum marianum* introduced into feed rations for broiler chickens decreased (by about 10%) daily weight gain and increased (by 4.5%) the feed conversion rate (FCR) in comparison to birds fed diets without milk thistle. The weight gain of chickens decreased after using 5% or 15% of milk thistle (*Silybum marianum*) oil cake in feed rations, as noted by Šťastník et al. [43].

A decreased carcass yield after adding *Silybum marianum* to broiler diets was observed by Schiavone et al. [46] and Šťastník et al. [43]. Šťastník et al. [43] using 5% or 15% of milk thistle seeds in feed rations observed a decrease in carcass yield by 3.86% and 4.22% of carcass yield compared to control chickens. A linear decrease in carcass yield accompanied by an increase in the share of milk thistle in the diets of birds was also obtained by Schiavone et al. [46]. On the other hand, an increase (*p* < 0.05) in the dressing percentage of chickens after adding *Silybum marianum* in the amount of 15 g/kg of feed rations was observed by Ahmad et al. [20].

In the production of broilers breast muscle weight in relation to carcass weight is of economic significance. Breast muscles account for about 30% of edible meat in the whole carcass [72], which is corroborated by the results of own studies. The absence of any impact of *Silybum marianum* in broiler chicken diets on the share of breast muscles in the carcass is consistent with the findings of Šťastník et al. [43] and Rashidi et al. [49]. Rashidi et al. [49] also demonstrated that a share (0.5%, 1%, 1.5% and 2%) of *Silybum marianum* in diets had no impact on the share of abdominal fat. A significant decrease in the share of abdominal fat in the carcasses of birds fed with rations containing milk thistle extract (250 mg/kg) was found by Tavakolinasab et al. [7].

Šťastník et al. [43] reports that broilers receiving feed rations with a higher (15%) content of milk thistle had a higher (2.69% vs. 2.3%) share of liver than those fed with diets free from and containing 5% of *Silybum marianum*. Similarly, Tavakolinasab et al. [7] report that introducing milk thistle extract in the amount of 250 mg/kg of chicken diet leads to an increase (by 0.21%) in the share of liver.

Poultry meat available on the market should be of proper quality, perceived by consumers as a collection of many features, including nutritional value as one of the most important characteristics [73,74,75]. Poultry meat is a source of complete animal protein that—according to FAO/WHO (Food and Agriculture Organization of the United Nations/World Health Organization)—is equivalent to milk protein. In addition, its energy value is lower [76,77].

The absence of an impact of milk thistle in broiler chicken feed on the total protein content in breast and leg muscles corroborates the results obtained by Šťastník et al. [48]. However, own studies showed a significant reduction in the amount of crude fat contained in the leg muscles of birds receiving feed rations with milk thistle in comparison to the control group. Schiavone et al. [46] studied chemical properties of breast meat and showed that the application of 40 and 80 ppm dried extract of milk thistle fruit in broiler feed (1.19% and 1.74% in fresh weight, respectively) significantly reduced fat content (2.15%). Thigh fat decreased in birds fed 40 ppm dried *Silybum marianum* extract (3.81% live weight) compared to a control group (4.79% live weight) and birds fed 80 ppm dried SM fruit extract (4.22% live weight). Dry matter, protein and ash content of breast and thigh meat did not significantly differ between the control and dried *Silybum marianum* extract fed groups. In turn, Grela et al. [21] analyzing the content of crude fat in the *longissimus lumborum* muscle of fatteners fed with rations containing 3% or 6% of milk thistle seeds noted a significant reduction in the content of this ingredient only when the content of milk thistle in the diet was reduced.

Own studies showed an increased share of stearic and linoleic acids in the muscles of chickens receiving feed rations containing milk thistle seeds. In turn, Kralik et al. [47], using feed rations with 3% of milk thistle oil for slaughter chickens, noted a decreased share of those acids in the breast and leg muscles. On the other hand, Grela et al. [21], having introduced 3% or 6% of milk thistle seeds in the diets of fatteners, noted a significant decrease in the share of stearic acid and an increased content of linoleic acid in the *longissimus lumborum muscle*. The increased share of PUFA in the muscles of birds fed with diets containing *Silybum marianum* is contrary to the findings of Kralik et al. [47] but consistent with those of Grela et al. [21] in studies involving fatteners. In addition, the values of atherogenicity index (AI), thrombogenicity index (TI) and hypo- and hypercholesterolemic ratio (h/H) in the breast muscles of chickens receiving experimental feed rations corroborated the tendency noted in the studies by Grela et al. [21].

The studies and experiments carried out by Kralik et al. [47] and Šťastník et al. [48] showed different impacts of diets containing milk thistle on the acidity of broiler chickens’ meat. The results did not corroborate an impact of the nutrition used on the reaction of breast and leg muscles, while Kralik et al. [47] found a decrease in the initial pH and an increase in the final pH of breast muscles after adding 3% of oil to the diet. In turn, Šťastník et al. [48] noted an increased pH of breast muscles after adding 5% or 15% (of *Silybum marianum)* to broiler chicken feed rations.

Color is an important attribute taken into account by consumers when buying meat, and an important element of evaluating meat dishes during their consumption [78]. Meat of darker color due to a higher share of oxidized myoglobin, is less desired by consumers [75]. Broiler meat color depends on the genotype and age [79], animal technology conditions, and feeding regime [80]. Numerous studies [75,81,82] showed that the higher the pH of meat is, the darker its color and vice versa. Extremely high pH leads to DFD and low to PSE meat defect [83].

The obtained L color values in the evaluated muscles were characteristic of normal muscles, since Van Laack et al. [84] classified breast muscle tissue as normal (CIE L* ≤ 55.0) and lighter than normal (CIE L* > 60.0). A lighter L* color (61.25 and 62.58) and the yellow saturation (13.19 and 13.51) of breast muscles immediately after using 5% and 15% of *Silybum marianum* was observed by Šťastník et al. [48]. In turn, red saturation (4.86 and 5.16) corresponded to own results for breast muscles of birds fed with rations containing ground milk thistle seeds. In turn, Kralik et al. [47], having added milk thistle oil to the feed rations, demonstrated a lower (1.66) value for parameter a* and a higher (12.01) value for parameter b* in the breast muscles compared to the results of own studies.

The WHC results obtained for muscles of chickens fed with rations containing *Silybum marianum* point to an impairment of the water holding capacity, but Kralik et al. [47] noted a positive impact of milk thistle seeds on drip loss.

In these studies, a positive impact of diets containing milk thistle seeds on the organoleptic (sensory) characteristics of muscles was noted. However, Šťastník et al. [48], having introduced 5% or 15% of milk thistle into chicken diets, demonstrated a deteriorated flavor score for both types of muscles, although the difference was confirmed to be statistically significant only in the breast muscles of chickens receiving 5% *Silybum marianum* in diets as compared to other groups.

Higher notes for respective sensory characteristics of thigh muscles should be associated with a higher content of fat in comparison to breast muscles. Komprda et al. [85] underlined that leg meat contains more fat and flavour substances, and is thus a preferred consumer choice. According to Nowak and Trziszka [73], in selecting meat its palatability and nutritional value are equally important.

## 5. Conclusions

Based on the results of studies, the use of milk thistle seeds in broiler chicken starter/grower diets can be recommended in the amount of 2/3%. The introduction of *Silybum marianum* in both types of feed rations made it possible to obtain the highest body weight at the lowest feed conversion per body weight gain unit, without any influence on muscularity and fattening grade. At the same time, muscles of chickens fed diets containing the phytobiotic featured the healthiest fatty acid profile and good taste characteristics.

## Figures and Tables

**Figure 1 animals-11-01550-f001:**
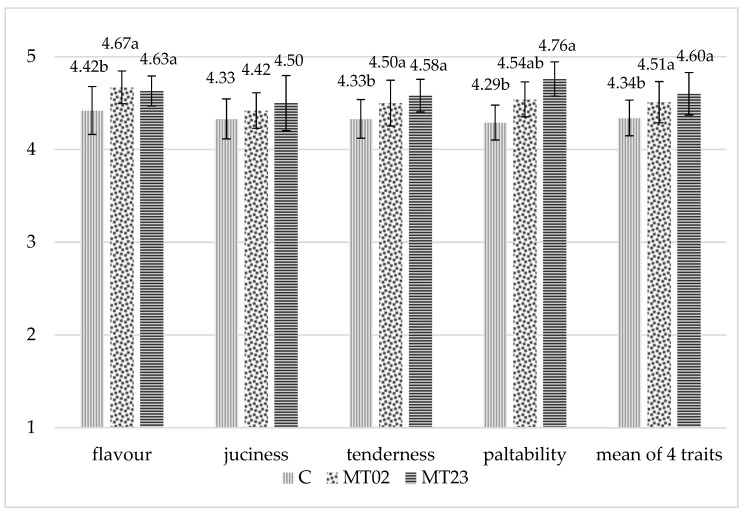
Sensory evaluation of breast muscles (point). C—control, MT02—milk thistle (0%/2%—starter/grower), MT23—milk thistle (2%/3%—starter/grower), SEM—standard error of mean, *n* = 8, a,b—means with different superscripts within a row are significantly different at *p* ≤ 0.05.

**Figure 2 animals-11-01550-f002:**
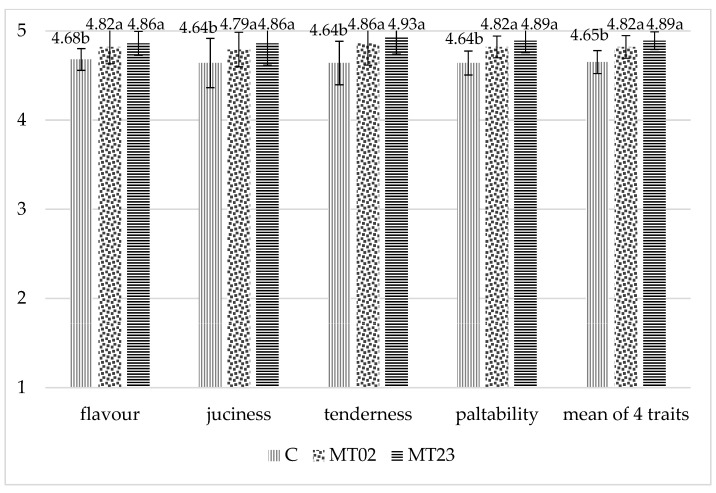
Sensory evaluation of thigh muscles (point). C—control, MT02—milk thistle (0%/2%—starter/grower), MT23—milk thistle (2%/3%—starter/grower), SEM—standard error of mean, *n* = 8, a,b—means with different superscripts within a row are significantly different at *p* ≤ 0.05.

**Table 1 animals-11-01550-t001:** Composition and nutritive value of rations.

Specification	Starter			Grower		
	C	MT02	MT23	C	MT02	MT23
Ingredients (g/kg)						
Wheat	598	598	618	608	628	618
Soybean cake	360	360	320	330	290	290
Milk thistle	-	-	20.0	-	20.0	30.0
Soybean oil	-	-	-	20.0	20.0	20.0
Limestone	11.5	11.5	11.0	10.8	11.7	11.7
NaCl	3.50	3.50	3.50	3.50	3.50	3.50
2–Ca phosphate	18.5	18.5	18.0	18.5	17.3	17.3
Premix starter/grower *	5.00	5.00	5.00	5.00	5.00	5.00
L–lysine (99%)	1.00	1.00	2.00	1.50	2.00	2.00
DL–methionine (99%)	2.50	2.50	2.50	2.70	2.50	2.50
Calculated nutritive value per 1 kg of diets						
Metabolizable energy (MJ)	13.0	13.0	13.0	13.3	13.4	13.4
Crude protein (g)	230	230	230	214	214	218
Lysine (g)	12.3	12.3	12.3	12.0	11.5	11.5
Methionine (g)	5.65	5.65	5.46	5.69	5.30	5.28
Tryptophan (g)	2.83	2.78	2.63	2.67	2.47	2.46
Ca total (g)	9.70	9.70	9.29	9.35	9.30	9.29
P available (g)	4.38	4.38	4.24	4.33	4.08	4.07
Na (g)	1.58	1.58	1.57	1.57	1.56	1.56
Analyzed nutrients (%)						
Dry matter	89.0	90.0	90.0	89.3	89.7	89.7
Crude ash	5.86	6.11	5.88	5.66	5.85	5.94
Crude protein	23.3	23.0	23.0	21.3	21.1	21.8
Crude fat	3.62	3.47	3.90	4.19	4.62	4.95
Crude fiber	2.98	3.12	4.13	3.26	4.25	4.19

* Mineral and vitamin starter/grower premix contained, per 1 kg; mg: choline chloride 140,000/80,000, Fe 16,000/14,000, Cu 4000/2400, Co 60, Mn 24,000/20,000, Zn 22,000/12,000, I 300/200, Se 40/50, antioxidants (butylated hydroxyanisole, butylated 99 hydroxytoluene); IU:2,800,000/2,000,000 vit. A, 100,000/600,000 vit. D3; mg: 14,400/10,000 vit. E, 800/600 vit. K3, 800/400 vit. B1, 1800/1400 vit. B2, 1200/800 vit. B6, 3000/2800 pantothenic acid, 12,000/6000 nicotinic acid, 400/300 folic acid, 60/30 biotin.

**Table 2 animals-11-01550-t002:** Basic nutrients, energy value and fatty acids profile of milk thistle seeds.

Specification	Composition (*n* = 3)
Basical nutrients (g/kg)	
Dry matter	886
Crude ash	31.5
Crude protein	219
Crude fat	238
Crude fiber	41.3
N-free extractives	356.2
Gross energy value (kcal/kg)	4145
Fatty acids (% total FA)	
C12:0	0.010
C14:0	0.110
C16:0	8.52
C16:1	0.090
C18:0	4.78
C18:1	23.59
C18:2	54.64
C18:3	0.160
C20:0	3.25
C20:1	0.920
C22:0	2.41
C22:4	0.190
others	1.33
SFA	19.08
UFA	79.59
MUFA	24.60
PUFA	54.99
DFA (UFA + C18:0)	84.37
OFA (C14:0 + C16:0)	8.63
AI	0.110
TI	0.330
h/H	9.11

FA—fatty acids, SFA—saturated fatty acids, UFA—unsaturated fatty acids, MUFA—monounsaturated fatty acids, PUFA—polyunsaturated fatty acids, DFA—neutral and hypocholesterolemic fatty acids, OFA—hypercholesterolemic fatty acids, AI—atherogenicity index, TI—thrombogenicity index, h/H—hypocholesterolaemic/Hypercholesterolaemic ratio.

**Table 3 animals-11-01550-t003:** Rearing results of broiler chickens.

Indicators	Groups			SEM	*p*–Value
	C	MT02	MT23		
Body weight (g)					
1 d	39.8	39.3	39.2	0.083	0.051
21 d	667	666	674	3.56	0.680
42 d	2306 ^b^	2377 ^ab^	2389 ^a^	14.75	<0.05
Body weight gain (g)					
1–21 d	627	627	635	3.59	0.657
22–42 d	1639	1711	1715	15.10	0.056
1–42 d	2267 ^b^	2338 ^ab^	2350 ^a^	14.76	<0.05
Feed conversion ratio (kg)					
1–21 d	1.61 ^a^	1.57 ^a^	1.51 ^b^	0.013	<0.05
22–42 d	1.85 ^a^	1.79 ^b^	1.71 ^c^	0.016	<0.05
1–42 d	1.74 ^a^	1.69 ^b^	1.61 ^c^	0.015	<0.05

C—control, MT02—milk thistle (0%/2%—starter/grower), MT23—milk thistle (2%/3%—starter/grower), SEM—standard error of mean, *n* = 5, chickens survivability = 100%, abc—means with different superscripts within a row are significantly different at *p* ≤ 0.05.

**Table 4 animals-11-01550-t004:** Slaughter analysis of broiler chickens.

Parameters	Group			SEM	*p*–Value
	C	MT02	MT23		
Body weight before slaughter (g)	2330	2392	2370	28.4	0.069
Cold carcasses weight (g)	1839	1875	1856	28.2	0.882
Dressing percentage (%)	78.9	78.4	78.2	0.532	0.888
Muscles total (%)	51.4	51.4	50.1	0.472	0.466
breast	30.0	29.9	29.7	0.436	0.975
thigh	12.7	12.4	12.3	0.166	0.543
drumstick	8.74 ^ab^	9.03 ^a^	8.13 ^b^	0.139	<0.05
Abdominal fat (%)	0.62	0.79	0.82	0.040	0.098
Skin with subcutaneous fat (%)	7.38	7.31	7.68	0.167	0.659
Giblets total share in body weight before slaughter (%)	3.30	3.20	3.36	0.029	0.299
heart	0.50	0.48	0.46	0.010	0.305
liver	1.79	1.61	1.71	0.036	0.124
stomach	1.01	1.11	1.18	0.030	0.056

C—control, MT02—milk thistle (0%/2%—starter/grower), MT23—milk thistle (2%/3%—starter/grower), SEM—standard error of mean, *n* = 10, ab—means with different superscripts within a row are significantly different at *p* ≤ 0.05.

**Table 5 animals-11-01550-t005:** Basic nutrients (g/100 g) of muscles.

Specification	Group			SEM	*p*–Value
	C	MT02	MT23		
Breast muscles					
Dry matter	25.4	25.0	25.0	0.134	0.292
Crude ash	1.17 ^a^	1.12 ^b^	1.16 ^ab^	0.007	<0.05
Crude protein	22.3	22.2	22.0	0.094	0.576
Crude fat	1.13	1.10	0.92	0.639	0.399
Leg muscles					
Dry matter	24.5	24.5	24.1	0.212	0.726
Crude ash	1.06	1.07	1.07	0.004	0.405
Crude protein	18.8	19.4	19.5	0.144	0.102
Crude fat	4.01 ^a^	3.43 ^b^	3.25 ^b^	0.120	<0.05

C—control, MT02—milk thistle (0%/2%—starter/grower), MT23—milk thistle (2%/3%—starter/grower), SEM—standard error of mean, *n* = 5 (*n* = ♂ + ♀), ab—means with different superscripts within a row are significantly different at *p* ≤ 0.05.

**Table 6 animals-11-01550-t006:** Fatty acids profile (% total FA) of muscles.

Fatty Acids	Group			SEM	*p*–Value
	C	MT02	MT23		
Breast muscles					
C14:0	0.110	0.110	0.100	0.004	0.239
C16:0	22.19	21.71	21.45	0.231	0.285
C18:0	5.92	5.91	5.82	0.115	0.941
C18:1	34.33	33.84	32.19	0.433	0.065
C18:2 n–6	32.59 ^b^	33.87 ^ab^	36.00 ^a^	0.561	<0.05
C18:3 n–3	1.05 ^b^	1.25 ^ab^	1.49 ^a^	0.080	<0.05
C20:0	0.100	0.100	0.090	0.005	0.587
C20:1	0.070 ^b^	0.110 ^a^	0.050 ^b^	0.010	<0.05
C20:2	0.080	0.080	0.070	0.004	0.743
C20:3 n–3	0.050	0.050	0.050	0.001	0.913
C20:4 n–6	0.600 ^a^	0.450 ^b^	0.450 ^b^	0.037	<0.05
SFA	28.45	27.99	27.62	0.278	0.382
UFA	71.46	71.89	72.25	0.276	0.402
MUFA	37.09 ^a^	36.19 ^ab^	34.19 ^b^	0.528	<0.05
PUFA	34.37 ^b^	35.70 ^ab^	38.06 ^a^	0.619	<0.05
DFA (UFA + C18:0)	77.38	77.80	78.07	0.227	0.317
OFA (C14:0 + C16:0)	22.30	21.82	21.55	0.234	0.278
n–6:n–3	30.17 ^a^	26.40 ^ab^	23.67 ^b^	0.600	<0.05
AI	0.320	0.311	0.302	0.006	0.387
TI	0.731	0.712	0.682	0.018	0.849
h/H	3.08	3.18	3.26	0.108	0.989
Leg muscles					
C14:0	0.110	0.120	0.110	0.003	0.268
C16:0	21.41	21.66	21.24	0.091	0.156
C18:0	4.72 ^b^	4.94 ^b^	5.69 ^a^	0.140	<0.05
C18:1	35.46 ^a^	34.60 ^a^	32.05 ^b^	0.521	<0.05
C18:2 n–6	32.83 ^b^	33.62 ^ab^	35.63 ^a^	0.476	<0.05
C18:3 n–3	1.65	1.42	1.61	0.054	0.174
C20:0	0.070 ^b^	0.080 ^ab^	0.130 ^a^	0.009	<0.05
C20:1	0.100 ^a^	0.060 ^b^	0.040 ^b^	0.008	<0.05
C20:2	0.030 ^b^	0.030 ^b^	0.060 ^a^	0.004	<0.05
C20:3 n–3	0.030	0.020	0.020	0.002	0.113
C20:4 n–6	0.320 ^a^	0.170 ^b^	0.310 ^a^	0.027	<0.05
SFA	26.40 ^b^	26.94 ^ab^	27.35 ^a^	0.161	<0.05
UFA	73.45 ^a^	72.94 ^ab^	72.46 ^b^	0.165	<0.05
MUFA	38.59 ^a^	37.68 ^a^	34.83 ^b^	0.577	<0.05
PUFA	34.86 ^b^	35.26 ^ab^	37.63 ^a^	0.491	<0.05
DFA (UFA + C18:0)	78.17	77.88	78.15	0.083	0.287
OFA (C14:0 + C16:0)	21.52	21.78	21.35	0.093	0.152
n–6:n–3	19.73	23.46	22.05	0.083	0.150
AI	0.301	0.300	0.300	0.006	0.365
TI	0.642	0.670	0.671	0.012	0.984
h/H	3.27	3.21	3.26	0.905	0.877

C—control, MT02—milk thistle (0%/2%—starter/grower), MT23—milk thistle (2%/3%—starter/grower), SEM—standard error of mean, *n* = 5, ab—means with different superscripts within a row are significantly different at *p* ≤ 0.05, SFA—saturated fatty acids, UFA—unsaturated fatty acids, MUFA—monounsaturated fatty acids, PUFA—polyunsaturated fatty acids, DFA—neutral and hypocholesterolemic fatty acids, OFA—hypercholesterolemic fatty acids, AI—atherogenicity index, TI—thrombogenicity index, h/H—hypocholesterolaemic/Hypercholesterolaemic ratio.

**Table 7 animals-11-01550-t007:** Physical parameters of muscles.

Parameters	Group			SEM	*p*–Value
	C	MT02	MT23		
Breast muscles					
pH_15_	6.27	6.17	6.30	0.315	0.218
pH_24_	5.60	5.92	5.85	0.060	0.058
L*	53.4 ^a^	49.4 ^b^	50.8 ^ab^	0.589	<0.05
a*	2.90 ^b^	5.17 ^a^	4.46 ^a^	0.284	<0.05
b*	0.810	0.790	1.19	0.168	0.579
C = [(a*)^2^ + (b*)^2^]^0.5^	3.08 ^b^	5.29 ^a^	4.71 ^a^	0.280	<0.05
H = log(b*/a*)	0.290	0.140	0.300	0.046	0.327
WHC (%)	7.46 ^b^	11.0 ^a^	12.7 ^a^	0.970	<0.05
Thigh muscles					
pH_15_	6.02	6.20	6.05	0.038	0.114
pH_24_	6.00	5.93	5.98	0.033	0.659
L*	50.7	51.5	53.2	0.583	0.190
a*	3.34	2.64	2.72	0.290	0.591
b*	0.860	0.390	0.510	0.248	0.740
C = [(a*)^2^ + (b*)^2^]^0.5^	3.80	2.99	2.86	0.278	0.342
H = log(b*/a*)	0.320	0.135	0.184	0.106	0.773
WHC (%)	5.50	5.56	6.97	0.637	0.644

C—control, MT02—milk thistle (0%/2%—starter/grower), MT23—milk thistle (2%/3%–starter/grower), SEM—standard error of mean, *n* = 10, ab—means with different superscripts within a row are significantly different at *p* ≤ 0.05, L*—lightness, a*—redness, b*—yellowness, C—chroma, H—hue, WHC—water holding capacity.

## Data Availability

The data are available on request from the corresponding author.

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
