# Peer review of "Impact of Milk Thistle (Silybum marianum [L.] Gaertn.) Seeds in Broiler Chicken Diets on Rearing Results, Carcass Composition, and Meat Quality"

_animals, 2021, doi:10.3390/ani11061550_

Round 1
Reviewer 1 Report
This manuscript by Janocha and colleagues have determined the effect of dietary milk thistle (Silybum marianum [L.] Gaertn.) seeds supplementation on rearing results, carcass composition and meat quality of broilers. This is an interesting topic. The manuscript could be published after the following revision.
- Abstract, Lines 19-20,the experimental design did not explained very well. Please add the replicates information in the main text.
- Line 42, “P < 0.05”, please note that “P” needs to be wrote in italic, the space needs to be added before and after “<”. Please check this throughout the paper.
- Please add the degree of the changes (percentage of the changes) in the results section in the abstract.
- No conclusion in the Abstract?
- Please add some recently references (2018-now) about the functions of milk thistle extract.
- Please add some references about the nutrition values evaluation of milk thistle seeds. Not only the function of milk thistle extract or their function compounds.
- Please add the rational of the added doses of milk thistle seeds in the broiler diets.
- Please check the amount of the “Crude protein (g)” of MT02 in the starter period in Table 1.
- Meanwhile, in Table 1, how do you calculate the “Metabolizable energy (MJ)” values for the diet contained milk thistle seeds? Do you have the nutrition values of milk thistle seeds? Please add the analyzed nutrition value data of milk thistle seeds in the main text as the Table 1.
- Please add the replicates information in the footnotes of all the Tables.
- Please check the figure legends of Figures 1 and 2. Important information missed. Such as Values expressed as mean ± SE or SD? Replicates n=? P value set for the significant changes?
- Lines 405-406, no “p<0.05” values need to be written in the discussion or conclusion section.
Author Response
Response to Reviewer 1 Comments
- Abstract, Lines 19-20,the experimental design did not explained very well. Please add the replicates information in the main text.
Response 1: The replicates information in the main text of abstract were added.
- Line 42, “P < 0.05”, please note that “P” needs to be wrote in italic, the space needs to be added before and after “<”. Please check this throughout the paper.
Response 2: “P” in italic were written. The space before and after “<” were added. This was checked throughout the paper.
- Please add the degree of the changes (percentage of the changes) in the results section in the abstract.
Response 3: Percentage of the changes in the results section in the abstract were added.
- No conclusion in the Abstract?
Response 4: See the last sentence, please.
- Please add some recently references (2018-now) about the functions of milk thistle extract.
Response 5: Recently references about the functions of milk thistle extract was added.
- Please add some references about the nutrition values evaluation of milk thistle seeds. Not only the function of milk thistle extract or their function compounds.
Response 6: References about the nutrition values evaluation of milk thistle seeds was added.
- Please add the rational of the added doses of milk thistle seeds in the broiler diets.
Response 7: It was added.
- Please check the amount of the “Crude protein (g)” of MT02 in the starter period in Table 1.
Response 8: There was a comma mistake. It was corrected.
- Meanwhile, in Table 1, how do you calculate the “Metabolizable energy (MJ)” values for the diet contained milk thistle seeds? Do you have the nutrition values of milk thistle seeds? Please add the analyzed nutrition value data of milk thistle seeds in the main text as the Table 1.
Response 9: The “Metabolizable energy (MJ)” values for the diets were calculated according to the European Tables. The analyzed nutrition value data of milk thistle seeds in the main text as the Table 1. was added.
- Please add the replicates information in the footnotes of all the Tables.
Response 10: The replicates information in the footnotes of all the Tables were added.
- Please check the figure legends of Figures 1 and 2. Important information missed. Such as Values expressed as mean ± SE or SD? Replicates n=? P value set for the significant changes?
Response 11: Legends of Figures 1 and 2 were supplied.
- Lines 405-406, no “p<0.05” values need to be written in the discussion or conclusion section.
Response 12: It was removed.
Sincerely
Anna Milczarek

Reviewer 2 Report
This manuscript try to assess the effect of feeding differente amounts of milk thistle seeds in different productive phases (starter and finisher) on broiler productive performance, carcass composition, meat quality and sensory properties. The article is interesing for the Journal audience but some points need to be addressed before I can consider this manuscript acceptable for publication. Therefore my decision is to reconsider after major revision.
Major concerns
- The introduction needs to be improved. I miss a hypothesis of what is the advantage/disadvantage of feeding milk thistle during the starter phase. In addition the objective is very vague considering that authors formulated different concentrations of milk thistle in different production phases (starter vs finisher). Otherwise there is a lack of coherence between the introduction and the experimental design. I also miss a reason explainining the contradictory results found in the literature which could highlight the novelty of the current trial.
- Authors have analyzed the seed to determine the FA profile. However, what was fed to birds was a concentrate which was formulated with soybean oil (2% during the finisher period). In order to make results and implications clearer I suggest reporting the FA profile of the concentrates.
- Experimental unit definition. Authors need to clearly indicate what have considered as experimenal unit. The most appropriate experimental unit is the smallest unit upon which a treatment can be applied. If the seeds were formulated in the concentrate, and birds were group fed in the cage, then I presume it should be the cage not the individual animal.
- Statistical analysis. For treatments that are fed to groups of animals in pens ,and all animals are treated similarly, then the statistical advantage gained by measuring responses for groups of animals within pen, rather than one animal, is to achieve a more precise estimate of the value for that pen. Authors have sampled 10 birds per treatment, but authors should have averaged individual measurements to the pen (the experimental unit). Please, clearly state how these 10 birds have been selected from 5 replicates and how they have been treated previous to the statistical analysis. Authors cannot use in the statistical analysis more than 5 values per measured phenotype.
- Post hoc test. Authors used a Duncan test as post hoc test to differentiate treatment effects. This is not correct. It does not protect against the familywise error rate so can lead to overstimation of the type I error. Since authors perform pairwise comparisons between 3 different treatments I suggest reanalyzing data using a more protective post hot test: tukey, Bonferroni for instance.
- Results, discussion and conclusion sections should be reworded according to the new statistical analysis.
Minor comments
Line 15"..in both types of feed rations". It is not clear what authors mean. Please reword.
Line 22 "..ground seeds". In the material and methods section it is not mentioned that seeds were grounded.
Line 23-24. According to what table? Table 3 and 4 show contradictory results.
Line 30-31. Please be more specific. It is not clear what the effect of the different treatment is.
Line 52 and 54. Add a reference
Line 59. Please reword
Line 90. Did author use sexed birds?
Line 95. Is this the standard protocol for rearing Ross broilers? Was this protocol approved by an ethical committee?
Line 99. Soybean oil was not formulated during the starter phase. Did authors formulate an enzyme to help digest wheat starch?
Line 101. Grounded seed? what was the form of the concentrate?
Table 1. Delete 1 kg of the table heading. Clearly indicate the units instead (Please use g/kg). Report if it is on DM or as fed basis.
Use of decimals in Tables: use 3 decimals with means below 0, 2 decimals with means between 1 and 10, 1 decimal with means between 11 and 100, and do not use decimals with means greater tan 100. Provide SEM values with one decimal more than that used for the lsmean. Define all the abbreviatons used in table footnote.
Line 113-114. Please check major concern n3. Indicate how birds were slaughtered. Please provide the ethical approval code.
Line 113 "...birds with a body weight representative...". When all animals were weighted significant differences were found among treatmens (Table 3) but non significant differences were found when authors tried to take a representative sampling. Would it not have been better to take them randomly?
Line 114-119. This is not experimental design.
Line 122-125. Information on how brutto energy, crude fibre and N-free extractives were calculated is missing.
Line 126. Report how samples for seed analysis were taken. This information is missing.
Line 171. Avoid using terms such as advantageous in the results section.
Table 2. Describe the seed sampling procedure. In the discussion section a range of values is reported but a SD (or even the range) value is missing in this table. What is brutto? gross energy? Soybean meal instead of soybean cake.
Table 3. Did authors measure mortality? If so, please report it and I suggest calculating the european production efficiency factor. Why use the SBM abbreviation for the control? All the concentrates were formulated with SBM.
Line 192. How can authors explain a significant difference between the SBM and MT02 during the starter phase? According to table 1 these animals were fed the same concentrate.
Table 4. I suggest deleting body weight before slaughter.
Line 206-207. Be more specific in describing treatment effects.
Line 214-216. Please reword.
Table 6. I miss the n6:n3 ratio
Line 229-232. It is confusing. Please reword to clearly state that only MT23 resulted in significant differences compared to the other treatments.
Line 234-236. Please reword to make a comparison to the other treatments. Also report that MT02 did not show significant differences compared to SBM
Line 236-238. Please reword to make a comparison among treatments.
Author Response
Response to Reviewer 2 Comments
Major concerns
- The introduction needs to be improved. I miss a hypothesis of what is the advantage/disadvantage of feeding milk thistle during the starter phase. In addition the objective is very vague considering that authors formulated different concentrations of milk thistle in different production phases (starter vs finisher). Otherwise there is a lack of coherence between the introduction and the experimental design. I also miss a reason explainining the contradictory results found in the literature which could highlight the novelty of the current trial.
Response 1: The introduction according suggestion were improved. We would try a coherence between the introduction and the experimental design. Sadowska et al. [2011] claimed, that high content of cellulose and lignin fraction in the diets of monogastric animals can pose certain limitations. So we used different concentrations of milk thistle in different production phases (starter vs finisher). Probably, the contradictory results found in the literature we could explain a different forms (meal, cake, extract) and share of milk thistle in chicken rations were used.
- Authors have analyzed the seed to determine the FA profile. However, what was fed to birds was a concentrate which was formulated with soybean oil (2% during the finisher period). In order to make results and implications clearer I suggest reporting the FA profile of the concentrates.
Response 2: We analyzed the milk thistle seed to determine the FA profile. Unfortunately, we did not analyze the rations to determine the FA profile, so we could not give that.
- Experimental unit definition. Authors need to clearly indicate what have considered as experimenal unit. The most appropriate experimental unit is the smallest unit upon which a treatment can be applied. If the seeds were formulated in the concentrate, and birds were group fed in the cage, then I presume it should be the cage not the individual animal.
Response 3: The milk thistle were formulated in the diets, and birds were group fed in the cage, so a cage would be the experimental unit.
- Statistical analysis. For treatments that are fed to groups of animals in pens ,and all animals are treated similarly, then the statistical advantage gained by measuring responses for groups of animals within pen, rather than one animal, is to achieve a more precise estimate of the value for that pen. Authors have sampled 10 birds per treatment, but authors should have averaged individual measurements to the pen (the experimental unit). Please, clearly state how these 10 birds have been selected from 5 replicates and how they have been treated previous to the statistical analysis. Authors cannot use in the statistical analysis more than 5 values per measured phenotype.
Response 4: The information about statistical analysis treatment we gave in material and methods and beneath the tables of manuscript.
- Post hoc test. Authors used a Duncan test as post hoc test to differentiate treatment effects. This is not correct. It does not protect against the familywise error rate so can lead to overstimation of the type I error. Since authors perform pairwise comparisons between 3 different treatments I suggest reanalyzing data using a more protective post hot test: tukey, Bonferroni for instance.
Response 5: We corrected post hoc test. We used Tukey test.
- Results, discussion and conclusion sections should be reworded according to the new statistical analysis.
Response 6: Results, discussion and conclusion sections were reworded according to the new statistical analysis according Tukey test.
Minor comments
Line 15"..in both types of feed rations". It is not clear what authors mean. Please reword.
Response: Line 15. It was reworded “…in starter and grower rations…”
Line 22 "..ground seeds". In the material and methods section it is not mentioned that seeds were grounded.
Response: Line 22. We used "ground seeds". It was added in the material and methods section.Line 23-24. According to what table? Table 3 and 4 show contradictory results.
Line 23-24. According to what table? Table 3 and 4 show contradictory results.
Response: Line 23-24. According table 3. Table 3 and 4 do not show contradictory results. In the table 3 body weight in 42nd day was given, whereas in the table 4 body weight of chicken which were chosen to slaughter.
Line 30-31. Please be more specific. It is not clear what the effect of the different treatment is.
Response: Line 30-31. It was improve.
Line 52 and 54. Add a reference
Response: Line 52 and 54. There were added.
Line 59. Please reword
Response: Line 59. It was reworded.
Line 90. Did author use sexed birds?
Response: Line 90. Yes. It was added in material and method section of the manuscript.
Line 95. Is this the standard protocol for rearing Ross broilers? Was this protocol approved by an ethical committee?
Response: Line 95. It was added. See lines 475-480, please.
Line 99. Soybean oil was not formulated during the starter phase. Did authors formulate an enzyme to help digest wheat starch?
Response: Line 99. Soybean oil was not used during the starter phase. We used full-fat soybean cake. Enzyme was not add to diets.
Line 101. Grounded seed? what was the form of the concentrate?
Response: Line 101. Yes. Milk thistle seeds were grounded. Rations (starter and grower) were in loose form. It was added.
Table 1. Delete 1 kg of the table heading. Clearly indicate the units instead (Please use g/kg). Report if it is on DM or as fed basis.
Use of decimals in Tables: use 3 decimals with means below 0, 2 decimals with means between 1 and 10, 1 decimal with means between 11 and 100, and do not use decimals with means greater tan 100. Provide SEM values with one decimal more than that used for the lsmean. Define all the abbreviatons used in table footnote.
Response: Table 1. It were done.
Line 113-114. Please check major concern n3. Indicate how birds were slaughtered. Please provide the ethical approval code.
Response: Line 113-114. It was added. See lines 475-480, please.
Line 113 "...birds with a body weight representative...". When all animals were weighted significant differences were found among treatmens (Table 3) but non significant differences were found when authors tried to take a representative sampling. Would it not have been better to take them randomly?
Response: Line 113. Representative (average) body weight a group and sex birds to slaughter were chosen. It was given in material and methods section. In our opinion, it would not better to take chickens to slaughter randomly.
Line 114-119. This is not experimental design.
Response: Line 114-119. It was improved.
Line 122-125. Information on how brutto energy, crude fibre and N-free extractives were calculated is missing.
Response: Line 122-125. The information were added.
Line 126. Report how samples for seed analysis were taken. This information is missing.
Response: Line 126. The information was added.
Line 171. Avoid using terms such as advantageous in the results section.
Response: Line 171. It was improved.
Table 2. Describe the seed sampling procedure. In the discussion section a range of values is reported but a SD (or even the range) value is missing in this table. What is brutto? gross energy? Soybean meal instead of soybean cake.
Response: Table 2. It was gross energy and soybean cake.
Table 3. Did authors measure mortality? If so, please report it and I suggest calculating the european production efficiency factor. Why use the SBM abbreviation for the control? All the concentrates were formulated with SBM.
Response: Table 3. Chickens survivability was 100%. SBM abbreviation for the control was changed to “C”. All rations contained soybean cake.
Line 192. How can authors explain a significant difference between the SBM and MT02 during the starter phase? According to table 1 these animals were fed the same concentrate.
Response: Line 192. Statistical analyzes using Tukey test did not show a significant difference between the C (SBM) and MT02 during the starter phase.
Table 4. I suggest deleting body weight before slaughter.
Response: Table 4. We used the body weight before slaughter to calculate dressing percentage of birds.
Line 206-207. Be more specific in describing treatment effects.
Response: Line 206-207. It was improved.Line 214-216. Please reword.
Line 214-216. Please reword.
Response: Line 214-216. It were changed.
Table 6. I miss the n6:n3 ratio
Response: Table 6. The n6:n3 ratio were calculated and added.
Line 229-232. It is confusing. Please reword to clearly state that only MT23 resulted in significant differences compared to the other treatments.
Response: Line 229-232. It was reworded.
Line 234-236. Please reword to make a comparison to the other treatments. Also report that MT02 did not show significant differences compared to SBM
Response: Line 234-236. It was changed.
Line 236-238. Please reword to make a comparison among treatments.
Response: Line 236-238. It was changed.
Sincerely
Anna Milczarek

Round 2
Reviewer 1 Report
No further comments.
Reviewer 2 Report
Authors have adquatly addressed all the raised questions. Therefore, my suggestion is to accept the manuscript in the current form.